# Men's late presentation for HIV care in Eastern Uganda: The role of masculinity norms

Sherifah Nabikande[1]*, Juliana Namutundu[1], Joanita Nangendo[2], Tom Okello[2], Winnie Agwang[1], Joan Tusabe[1], Steven Ndugwa Kabwama[1,3], Anne Ruhweza Katahoire[4]

1 School of Public Health, College of Health Sciences, Makerere University, Kampala, Uganda, 2 Clinical Epidemiology Unit, College of Health Sciences, Makerere University, Kampala, Uganda, 3 Uganda Public Health Fellowship Program, Ministry of Health, Kampala, Uganda, 4 Child Health and Development Centre, College of Health Sciences, Makerere University, Kampala, Uganda

* nabikandesharifah@live.com

## Abstract

### Introduction

In Uganda, adult men living with HIV are more likely to present late for care; with a CD4 cell count below 350 cells/μl compared to women. Understanding why adult men present late for HIV care is important in improving early linkage to care. Studies across countries in Sub-Saharan Africa emphasize the role of masculinity norms; defined as social expectations about appropriate roles and behavior for men, in men's health behaviours particularly, in HIV care engagement. This study therefore explored how masculinity norms influence men's late presentation for HIV care.

### Methods

This was a qualitative study undertaken in Jinja District, Eastern Uganda between October and November 2020. We conducted 20 In-Depth Interviews (IDIs) with men living with HIV who had presented late for care at Family Hope Centre. We also conducted four Focus Group Discussions (FGDs) with HIV negative men and women in selected communities of Katende and Walukuba. Conventional content analysis approach was used to identify themes across the collected data.

### Results

A total of 20 men participated in the In-depth Interviews (IDIs), with majority being married 15/20 (75%) and primary level holders 7/15 (46.7%). Nineteen (19) women participated in two FGDs, with a mean age of 29.5 years. Nineteen (19) men also participated in other two FGDs, with a mean age of 28.2 years. Conventional content analysis results indicated that men's late presentation for HIV care in Jinja district is greatly related to their concerns of loss of respect and the need to preserve their reputation and maintain a sense of normality in their families and society as proposed by Wilson's (1969) respectability-reputation theoretical model. Respectability was endorsed by 'the wider society', while reputation was endorsed almost entirely by men and some women.

**Data Availability Statement:** All relevant data are within the paper and its Supporting Information files.

**Funding:** Research reported in this publication was supported by the Fogarty International Center, National Institute of Alcohol Abuse and Alcoholism, National Institute of Mental Health, of the National Institutes of Health under Award Number D43 TW011304. The content is solely the responsibility of the authors and does not necessarily represent the official views of the National Institutes of Health. SN received the funding. The funders had no role in the study design, data collection and analysis, decision to publish, or preparation of the manuscript.

**Competing interests:** The authors have declared that no competing interests exist.

## Conclusion

Findings show that the explanations for men's late presentation arise from the masculinity norms in Jinja District, Eastern Uganda.

## Introduction

Although women in most parts of Sub-Saharan Africa (SSA), and Uganda in particular, have a higher prevalence of HIV than men [1, 2], men living with HIV are more likely to present late for care (with a CD4 cell count below 350 cells/μl) compared to women [3–5]. The recent Uganda Population based HIV Impact Assessment (UPHIA) reports that 29.6% of adults newly diagnosed with HIV had a CD4 count of less than 350 cells/μl; slightly more than a third of these (35.1%) were men [2]. Fewer men seek testing and counselling services on a routine basis leading to late diagnosis in late-stage disease progression [6]. Low perceptions or knowledge about HIV/AIDS and the benefits of Voluntary Counseling and Testing (VCT) services have been reported for men's negative attitudes to access HIV/AIDS services [7]. The provision of information can improve individuals' HIV-related health literacy [8]. Health literacy is required in making critical health decisions to manage individuals' health, including to seek and access appropriate health care [9]. In addition, men's supportive social relationships with the healthcare professionals, their ease of access to the HIV care services, and their ability to seek the services have been reported to determine their access to the services [10]. Due to late presentation, men develop opportunistic infections with poor treatment outcomes [11] and high healthcare costs. The costs of medical treatment, such as the costs for standard blood, viral load and CD4 tests prior to the initiation of treatment, the treatment of opportunistic infections and administration have also been revealed to be barriers to the access of HIV care services among People Living with HIV (PLHIV) [10, 12–14]. This increases the risk of onward transmission due to uncontrolled viremia and lack of awareness of HIV status [15]. Early presentation for care is important for reducing the risk of mortality and improving quality of life among people with HIV [16]. It can also reduce the risks of virological failure, and loss to care (attrition) [17–19].

Masculinity norms are social expectations about appropriate roles and behavior for men [20]. These have been recognized as an important influence on men's health behaviours particularly, in HIV care engagement serving as both barriers and facilitators [21]. For example, while some norms emerge as facilitators to care for men who believe antiretroviral treatment (ART) will allow them to reassume masculine roles and relationships, these same norms serve as barriers to care for other men who fear that HIV will threaten their traditional roles, as fathers and husbands [21].

Earlier studies conducted in central and rural eastern Uganda among male fisher folk and those in the gold mining community respectively, highlighted the influence of masculinity norms on men's engagement in HIV care [22–24]. Among fishermen and other male fisherfolk, it was reported that the masculine roles of worker and provider, husband and sexual partner, and the appearance of physical strength were compromised by HIV, but restored by ART's positive effects on health yet occupational and structural factors specific to men's role as worker remained as barriers to engagement in HIV care [22]. In an artisanal gold mining community, men's fear to be, or actually being, judged as unable to work and being marginalized from work led many not to initiate HIV treatment or to drop out and this undermined their masculine repute as hard workers, earners and providers for their families [23]. Lastly, Siu and

colleagues report that men's treatment seeking behaviours corresponded with different masculine ideologies. Family and societal expectations to be a family provider and respectable role model encouraged treatment, to regain and maintain health [24]. However, reputational concern with strength and the capacity for hard physical work, income generation and sexual achievement discouraged uptake of HIV testing and treatment since it meant acknowledging weakness and an 'HIV patient' identity [24].

However, these studies were conducted among specific sub-populations whose social context did not necessarily reflect that of men in the general population irrespective of type of occupation. In addition, masculinities vary and the experience of masculinity is not uniform across or within cultural contexts. The central assertion here is that masculinity is dynamic, and enacted and sustained through and/or in different circumstances and as the society around a man changes through his lifetime. Hence it may vary according to the social groups of men that comprise a particular society [25]. This study therefore aimed to explore how masculinity norms influence men's presentation for HIV care in eastern Uganda.

## Theoretical framework

Masculine respectability and reputation. This study conceptualized masculinity from Wilson's (1969) concepts of 'respectability' and 'reputation'. Wilson's (1969) early ethnographic work suggests that these concepts inform two closely interconnected value systems by which men relate their position in society. These value systems structure men's social relationships, shape how their identities are produced, maintained, and challenged, and govern their conduct within the community [26].

Wilson states that "respectability" is the degree of conformity to the ideals of the whole legal society. By 'legal society,' he implied the moral values of institutions such as the family and church, in which one could participate in an official capacity. He argues that respectability accrued from and/or was affirmed by proper attention to the requisites of marriage and providing for children, consistent hard work, and adequate material possessions such as a home, economic independence and education, as well as the ideals of the church. Therefore, 'respectability' was concerned largely with morality and membership of, and active commitment to, the whole/external society [26].

On the other hand, "reputation" is the honour accrued to a man as a result of his 'masculine activities'. According to Wilson, reputation was almost entirely shaped by the perception of male peers and was oriented towards proficiency in all male activities and roles including sexual prowess, fathering many children, "gamesmanship" skills, including toughness and authority-defying behaviour. Wilson contends that reputation reflects the congruence of how a man views himself and how he is viewed by other males.

Wilson's respectability-reputation model provided a relevant framework for interpreting men's considerations regarding presentation for HIV care.

## Materials and methods

### Ethical considerations

We obtained ethical approval from Makerere University School of Public Health Research and Ethics Committee (FWA00011353). We also obtained administrative authorization from the Chief Administrative Officer, District Health Officer of Jinja district, and the management of Family Hope Centre. All study participants either gave written informed consent or used a thumbprint before interviews.

## Study setting and design

This study was conducted in Jinja District, Eastern Uganda between October and November 2020. Uganda is ethnically diverse, with over 30 different ethnic groups contained within its borders. Masculinity remains strongly tied to the breadwinner identity and one's role and way of participating in family and community life continues to be highly designated by one's sex. Men hold more power and status in both family and community spheres and are considered responsible for productive labor outside of the home. In addition, there exists one of the dominant masculine ideals of being a provider, primarily embodied through responsible fatherhood. Jinja District is a Patriarchal society in which men hold primary power and predominate in roles of political leadership, moral authority, social privilege and control of property. It is also patrilineal, meaning that property and title are inherited by the male lineage. The District has a population of about 502,815 people, with an HIV prevalence estimated at 4.7% among adults aged 15–64 years [2]. In 2019, approximately more than half (57%) of adult men newly initiated on ART in Jinja presented late for care [27]. The district has a number of HIV care sites that include; 1 Regional Referral Hospital, 2 PNFP Hospitals, 5 Health centre IVs, 3 special HIV clinics, 3 Armed forces clinics and 14 Health centre IIIs. This study was conducted in partnership with Family Hope Centre, a private health care facility in the district supported by the Children's Aids Fund Uganda (CAFU).

In this setting, a qualitative descriptive analysis situated within a constructionist epistemology was employed in exploring the role of masculinity norms in men's late presentation for HIV care.

## Study population, sampling and recruitment

The study was conducted among two populations; adult men living with HIV who presented late for care at Family Hope Centre and HIV negative men and woman in selected communities. Participants were eligible if; 1) were aged 18 years and above, 2) working or spending most of one's time in the selected communities and 3) willing to talk about masculinity norms and HIV. Exclusive criteria was inability to speak Lusoga or English and declining consent.

## Sampling and selection of study participants

We purposively sampled all study participants. Adult men living with HIV who presented late for care at Family Hope Centre were selected using the "maximum variation" strategy by including men of different age, education level, and marital status among others who had presented late for HIV care. On the other hand, HIV negative men and women in selected communities were selected using the "homogenous sampling" strategy by ensuring all participants in the same group are in the same age group and sex.

## Data collection procedures

Four (04) Research assistants fluent in both Lusoga and English with experience in qualitative data collection methods were recruited and trained for data collection.

A structured form was used to capture the basic socio-demographic information of the participants. In Depth Interviews (IDIs) among men living with HIV and Focus Group Discussions (FGDs) among HIV negative men and women were also used to collect data on the role of masculinity norms on men's late presentation for HIV care.

Before data collection, two pilot FGDs and one IDI were conducted in a village neighbouring the study area to test the flow of the discussions and interviews and how long each would last. Data was then collected using the IDI and FGD guides to explore the role of masculinity

norms in men's late presentation for HIV care in Jinja district. These elicited information on the beliefs and what it means to be a man in Jinja, and how this negatively affects men's presentation for HIV care. The study involved 20 IDIs and four FGDs (2 for women and 2 for men). The recruitment of participants ceased when this did not result in additional perspectives or information.

IDIs were held either in Lusoga or English depending on the participant's preference and interviews lasted between 30 to 40 minutes. Each FGD had 9–10 members and conversations were held in Lusoga and on average lasted one and a half hours. Discussions were guided by a trained moderator fluent in both English and Lusoga. A note taker also captured details of the FGDs.

With permission from all study participants, interviews and conversations were audio-recorded and recordings transferred to a computer and kept secure in Google drive for subsequent analysis.

## Data management and analysis

Three (3) transcribers fluent in Lusoga and English transcribed the audio interviews verbatim and typed into Microsoft Word with direct translation into English. In addition, since recruited Research Assistants were fluent in both Lusoga and English, translated transcripts were distributed among them to ensure correct interpretations.

The field research team and investigators read through all the transcripts and notes, then used open coding to derive codes from the text data and later developed categories and themes as indicated below "Fig 1". Data analysis was done manually. The analysis team comprised of three people (S.N, W.A and J.T) and these collaborated by holding physical team meetings throughout the analysis process. A week-by-week plan for analysis was drafted and the plan included a reference to the data analysis phase and coding assignments for each team member. During each iteration of coding, team members coded the same two to three transcripts into the codebook. During meetings, each coded statement was compared across members of the analysis team. Disagreements were discussed until the group reached consensus. Much of the discussions revolved around the similarities, differences, connections and patterns within and between the codes. The discussions revealed high levels of agreement between the team, and a consensus was able to be reached about the data. After all of the transcripts had been coded using consensus coding, the team then met one final time to review the codebook. During the meeting, the codebook was developed into a thematic structure that comprised of categories and associated themes that described the role of masculinity norms in men's presentation for HIV care. The themes were then reviewed and approved by all members of the analysis team.

Data analysis was guided by the five steps of qualitative data analysis framework [28]. These include (i) familiarization with the data by reading through transcripts, listening to the audios and going through the field notes; (ii) identification of a thematic framework by making notes, writing down recurrent themes and issues which emerge as important to study respondents; (iii) indexing the data by applying the thematic framework to the individual transcripts, grouping similar codes under the same themes and sub-themes; (iv) charting data through arranging appropriate thematic references in a summary of chart; and (v) mapping and interpretation of the data through which data were examined and interpreted [28, 29].

Procedures used to ensure trustworthiness of data and other processes. Investigators and research assistants had the required knowledge and skills to perform their roles. As mentioned under data collection procedures, two pilot FGDs and one IDI were conducted in a village neighbouring the study area to test the interview protocol. Investigators conducted regular debriefing sessions with research assistants and field notes were collected as referential

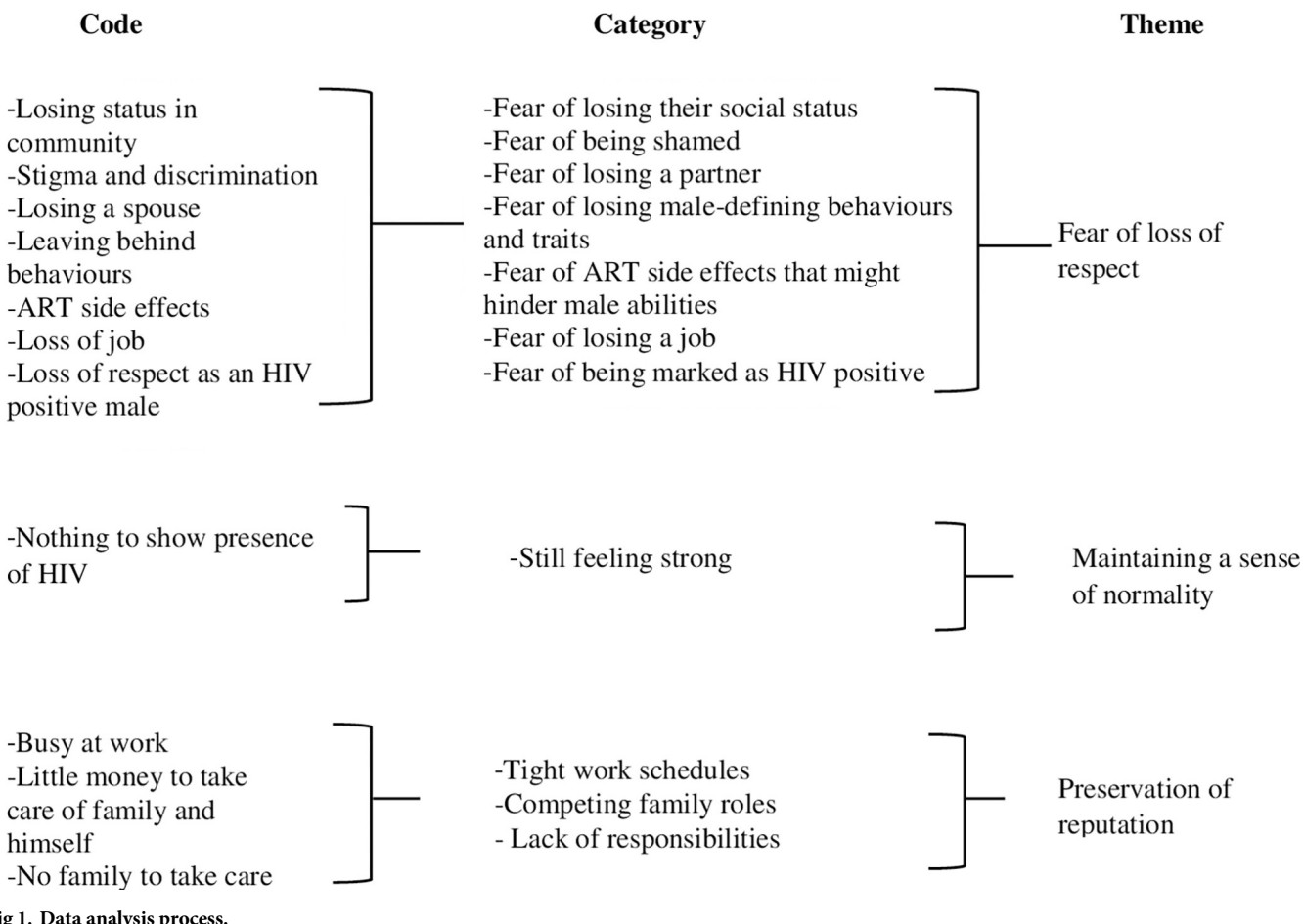

| Code | Category | Theme |
|------|----------|-------|
| -Losing status in community<br>-Stigma and discrimination<br>-Losing a spouse<br>-Leaving behind behaviours<br>-ART side effects<br>-Loss of job<br>-Loss of respect as an HIV positive male | -Fear of losing their social status<br>-Fear of being shamed<br>-Fear of losing a partner<br>-Fear of losing male-defining behaviours and traits<br>-Fear of ART side effects that might hinder male abilities<br>-Fear of losing a job<br>-Fear of being marked as HIV positive | Fear of loss of respect |
| -Nothing to show presence of HIV | -Still feeling strong | Maintaining a sense of normality |
| -Busy at work<br>-Little money to take care of family and himself<br>-No family to take care | -Tight work schedules<br>-Competing family roles<br>- Lack of responsibilities | Preservation of reputation |

**Fig 1. Data analysis process.**

materials during data analysis. Interviewers spent an average of 3–4 weeks to engage with study participants. We used methods triangulation to test validity through the convergence of information from both IDIs and FGDs. We also used two purposive sampling techniques to make sure that the selected participants were representative of the variety of views of adult men living with HIV who present late for care and HIV negative community members across Jinja.

## Results

### Characteristics of study participants

A total of 20 men participated in the In-depth Interviews (IDIs), with a mean age of 47.4 years. Majority were married 15/20 (75%) and primary level holders 7/15 (46.7%). A total of 38 individuals participated in the FGDs. Nineteen (19) women participated in two women's FGDs, with a mean age of 29.5 years. They were mainly housewives (n = 5) and businesswomen (n = 5). Nineteen (19) men also participated in two men's FGDs, with a mean age of 28.2 years. They were mainly casual workers (n = 7) (Table 1).

### Role of masculinity norms in men's late presentation for HIV care

Qualitative findings revealed three themes that highlighted the role of masculinity norms in men's late presentation for HIV care. These included; fear of loss of respect, preservation of reputation, and maintaining a sense of normality.

**Table 1. Characteristics of study participants.**

| IDI participants (all male, n = 20) | | FGD participants | | |
|---|---|---|---|---|
| | | n = 38 | | |
| Characteristic | Number | Characteristic | | Number |
| **Age group** | | age group | female | male |
| 18–39 | 6 | 18–39 | 15 | 18 |
| 40–49 | 4 | 40–49 | 3 | 1 |
| 50–75 | 10 | 50–57 | 1 | 0 |
| **Marital Status** | | Occupation | | |
| Single | 3 | student/not employed | 8 | 6 |
| Married/cohabiting | 15 | Professional# | 5 | 0 |
| Separated/widowed | 2 | Business/trading | 5 | 4 |
| **Highest education level attained** | | Others* | 1 | 9 |
| not educated/Primary | 8 | | | |
| Secondary | 8 | | | |
| Tertiary | 4 | | | |
| **Occupation** | | | | |
| Farming | 3 | | | |
| Professional# | 5 | | | |
| Business/trading | 6 | | | |
| Others* | 6 | | | |

#Professional; those practicing in a line of work after attaining formal qualification

*Others; boda rider (4/58), casual worker (7/58), carpenter (1/58), gardener (1/58), barber (1/58), tailor (1/58), forest preserver (1/58)

Fear of loss of respect. *Being shamed*. In almost all the IDIs and FGDs, men's fear of being shamed, stigmatized and discriminated against by the community was mentioned as a reason for late presentation for HIV care. An IDI participant was quoted saying:

> ". . .I feared that I will get ashamed thus did not start the medication. . .." (**IDI participant, 57 years**)

> "There is also stigma and also discrimination in my community and I feared people to discriminate against me." (**IDI participant, 32 years**)

*Being marked as HIV positive*. Participants mentioned that men feared to be seen at an ART clinic as this would serve as an indication that they were HIV positive. Men think that the medication is "like a belt tied on them for everyone to see." They fear people's perceptions of them and reactions once they are known to be positive. In particular, an IDI participant mentioned that a man thinks being seen as an HIV patient would reduce his love in the village:

> "He also thinks that the people are going to see him as an HIV patient when he starts the treatment. He would have reduced on love at the village and that is what most men fear." (**IDI participant, 58 years**)

*Losing a partner*. In some of the IDIs and FGDs, participants mentioned that men's fear of a spouse/partner leaving was also a barrier to presenting early for HIV care. In-laws of men living with HIV who were married often advise the wife to leave him before he dies. This is to prevent the woman from becoming a widow, since HIV was perceived to be fatal. An IDI participant reported to have hesitated presenting to HIV care for fear of his wife running away

from home. However, he met a health worker who told him to bring the wife to the facility for counselling.

"*I was worried that the wife was going to run away. There I got worried but I met a health worker called Rogers who told me that you bring her…….*" **(IDI participant, 25 years)**

"*The first problem is with the woman. Especially men fear their women. They think, "She will leave if she gets to know"* (**Adult women FGD1, Magwa Crescent)**

*Losing a job*. Both FGD and IDI participants mentioned that men delayed to present to HIV care fearing that their bosses or co-workers would find out about their HIV status and make them lose their jobs. It was reported that men are afraid of being labelled weak to engage in some work activities because they will end up being chased from work since they are no longer useful.

"*The nature of the jobs I do, in that if they come to know, they will say that I am weak so they don't want to give you hardships because I have witnessed many people who have been chased from work.*" (**IDI participant, 32 years**)

*Losing male-defining behaviors and traits*. IDI participants stated that men delayed to present for HIV care for fear of giving up leisure activities like alcohol drinking, smoking and making sexual advances which were a requirement for successful ART treatment.

"*Now there is someone who is being told that with this medicine you won't smoke, you won't take alcohol but when cigarette and alcohol is what he knows as his life…others because he is fearing that if they see me I will cease "vibing" girls…*" (**IDI participant, 45 years**)

Participants pointed out that some men delayed to present for HIV care due to fear of ART side effects. It was mentioned that men feared to lose some of their traits like; weakening due to the drug toxicity therefore unable to work and/or result into failure to father children.

"*People say that when you go and start the medication, some medicine stop men from producing…….That is why some people say that let me first produce before I go into treatment.*" (**IDI participant, 32 years**)

*Losing their social status*. It was mentioned that some men delay to start treatment because of fear of losing their social status, a problem an average man would not have. Those in high positions find it hard to start HIV care because of being worried about people talking about them. They are also worried that they will no longer be respected in society and anything they say will not be seen as relevant.

"*Some because of their social statuses. The average man wouldn't mind about what people talk about them. Those in high positions, I think those in high positions are normally reluctant.* (**IDI participant, 57 years**)

Preservation of reputation. *Tight work schedules*. FGD participants mentioned that men struggled to make time off work to report to the health facility for initiation. Main reasons were: some because of the nature of the work whereby someone is expected to report early in the morning and leave late in the evening when health workers are no longer at work as narrated by a male participant in FGD:

*"The work we do. You can have a job where you start work at 7 am and you will leave at 7pm, so that means you have no time to go and get medication. So I think that also causes a man not to get medication in time."* **(Adult men FGD2, Katende)**

*"Sometime a lot of work. Sometimes he says that I have a lot of work to do thus this also at times limits the man to go for treatment."* **(Adult women FGD2, Magwa Crescent)**

*Competing family roles.* It was also reported that men's role to care and provide for their families led men into neglecting their own health especially if their earnings weren't adequate enough yet medication was known not to be for free.

*"Because some people are there in the villages and they think that the medication is just sold. That also scares them because they know they have to look for some money for feeding at home and then look for money for medication…."* **(Adult men FGD2, Katende)**

*Lack of responsibilities.* In one FGD, it was reported that men who did not have any responsibilities for instance no family to look after were not motivated to start HIV care once they were diagnosed positive. Such men felt that they didn't have any value in the community and therefore no need to take drugs to prolong life.

*"You can be a man, true, but you do not have responsibility at home. You say to yourself. "I have no people I am looking after at home and now I am positive" in other words you will see yourself with no importance and you say to yourself "Why don't I die because there is no one that I am looking after?"* **(Adult men FGD1, Magwa Crescent)**

Maintaining a sense of normality. FGD participants reported that the absence of HIV symptoms at the time of diagnosis made men feel like the situation was not yet serious to warrant treatment or that the diagnosis was a false one. They added that some men first wait to experience body weakness and its only when they start seeing signs that guarantees being HIV positive. It's this that triggers them to wake up and start treatment at a later stage. Men who feel that they still have some energy think they will last without the treatment.

*"….what I think is that someone can delay if they have taken long to see the signs but yet they know they are positive and they tested. When they take long to see those signs they will say, "Aah, after all nothing shows. I am negative. Let me move on"* **(Adult men FGD2, Katende)**

*"I am fine that's when he doesn't have any sign on the body to show that he is sick. So he will have that in mind that I think they didn't check well and will first stay at that with false hope so when those weaknesses you know there is that body weakness that is caused by the disease and if there are some signs and symptoms that's when he will wake up and sometimes may wake up when it's a bit late……."* **(IDI participant, 40 years)**

## Discussion

This cross-sectional study assessed the role of masculinity norms in men's late presentation for HIV care. Key themes generated included; fear of loss of respect, preservation of reputation, and maintaining a sense of normality.

This study's findings show that men's late presentation for HIV care in Jinja district is greatly related to their concerns of loss of respect and the need to preserve their reputation and maintain a sense of normality in their families and society as proposed by Wilson's (1969) respectability-reputation theoretical model [26].

One of the most noticeable findings of this study was the fear of losing masculine respectability which can be seen in most of the sub-themes: fear of being marked as HIV positive, being shamed, losing a job, ART side effects and, losing a partner. The fear of being marked HIV positive and shamed was due to the explanation that being seen presenting at an HIV treatment center was seen as something that undermines a man's dignity in society. A study in Dar es Salaam, Tanzania [30] reported a similar finding, where it was found that men avoid going to the treatment clinics because doing so would amount to making one's HIV status known. Since society expects men to be wiser and exercise self-control in sexual decision making, being seen at an HIV clinic is interpreted as a failure in the man's sense of judgment, which undermines the respect and approval by others [24].

Different forms of stigma have been implicated as barriers to achieving the full potential of numerous clinical advances to improve health at the population level, including ART treatment as prevention [31–33]. Internalised stigma has been found to impact certain decisions like avoiding to seek care when needed [34]. Several studies have also shown an association between internal stigma and poor affective health and well-being as well as poor health care behaviors [35, 36]. Internalised/perceived HIV stigma encompasses feelings of being "less than" others, dirty or unclean, and/or deserving of negative outcomes due to having HIV [36]. A study in Indonesia reported that participants' individual moral judgement that associated their HIV status with amoral behaviours and engagement in sex with female sex workers, and participants' self-judgement as 'dirty people' or sinners which increased guilty feelings, were additional factors leading to HIV's perceived stigma [37]. On the other hand, "enacted stigma," which is the lived experience of discrimination against an HIV positive person through others' actions or words, is also known to have a negative impact on health outcomes [38]. Individuals have revealed having experienced HIV-related discrimination attitudes from healthcare professionals within healthcare settings. For example, in the Indonesia study, it was reported that clients were left untreated for many days in healthcare facilities [37]. HIV stigma and discrimination by healthcare professionals traumatized the PLHIV, thus concealment of HIV status, hindering access to healthcare services [37, 39, 40] thereby influencing their health and wellbeing in the long run.

In this study, men were also concerned about preserving their reputation by holding onto male defining behaviours such as; alcohol drinking, and smoking yet successful ART treatment required that they give these up. To men this meant losing a social life. An explanation for this is that one of the most important social aspects of masculinity that majority of men living with HIV lose is the freedom to enjoy the male dominated social spaces to interact with their fellow men [24]. The concern hindered their presentation for care early in the course of HIV infection. In line with this finding, some men reported avoiding or delaying the initiation of ART or clinic attendance for fear of having to change male sanctioned risk behaviours, including sexual risk (multiple partners and condom use) in Zimbabwe [41] and eastern Uganda [42], and alcohol use in South Africa [43].

This study's findings also show that men presented late for HIV care as a way of maintaining a sense of normality and preserving their reputation in society. It was reported that absence of HIV symptoms and still having strength at the time of diagnosis was one of the reasons for their late presentation to care. A number of studies found a similar finding. Siu and colleagues found that enduring physical symptoms of HIV and delaying treatment for as long as possible was viewed as a sign of strength and courage among some Ugandan miners [24]. Studies in Botswana and Uganda found that men in both rural and urban areas viewed seeking treatment as a sign of weakness [24, 41, 44], and men avoided care in South Africa because it made them feel vulnerable [45].

This study adds to the growing body of research evidence that masculinity norms have a role in men's access and engagement in HIV care. The generalizability of our findings is limited, given the qualitative nature of the study, as well as the use of purposive sampling to recruit participants. In addition, In Depth Interview (IDI) questions were based on past experiences and this might have resulted in recall bias. However, this was minimized by including participants who had presented for HIV care in recent months/years (between January 2018 and June 2020).

## Conclusion

Findings show that the explanations for men's late presentation arise from the masculinity norms in that context. These findings have some implications for Public Health interventions. HIV programs that seek to encourage early HIV testing and treatment need to be more attentive to the values upheld my men associated with how their societies and cultures expect them to behave, the attributes and responsibilities they should have to be considered as men in that context. For example, Antiretroviral Therapy could be promoted by emphasizing its value in rebuilding a man's respectability and role as head of family in regards to physical recovery. Supportive social spaces in which men can re-negotiate masculinities that are more aligned to their presentation for HIV care need to be created and men need to be supported with livelihood projects because this can help them provide for their families and encourage them to maintain HIV treatment. Further research is also needed to understand how masculinity norms can serve as a facilitator to men's presentation for HIV care.

## Supporting information

**S1 File.**
(DOCX)

**S2 File.**
(DOCX)

## Acknowledgments

Special thanks to the management and staff of Family Hope Center, Jinja for the help and support rendered to us during the whole data collection process. We also wish to acknowledge all study participants, the District Health Teams who enabled us to reach communities for data collection, the research assistants who ensured that we obtain good quality data, and the administrative teams who supported this study.

## Author Contributions

**Conceptualization:** Sherifah Nabikande, Juliana Namutundu, Steven Ndugwa Kabwama, Anne Ruhweza Katahoire.

**Formal analysis:** Sherifah Nabikande, Winnie Agwang, Joan Tusabe.

**Funding acquisition:** Sherifah Nabikande, Joanita Nangendo, Anne Ruhweza Katahoire.

**Investigation:** Sherifah Nabikande.

**Methodology:** Sherifah Nabikande, Juliana Namutundu, Steven Ndugwa Kabwama, Anne Ruhweza Katahoire.

**Project administration:** Sherifah Nabikande.

**Supervision:** Juliana Namutundu, Joanita Nangendo, Steven Ndugwa Kabwama, Anne Ruh-weza Katahoire.

**Validation:** Juliana Namutundu, Steven Ndugwa Kabwama, Anne Ruhweza Katahoire.

**Writing – original draft:** Sherifah Nabikande, Joanita Nangendo.

**Writing – review & editing:** Sherifah Nabikande, Juliana Namutundu, Joanita Nangendo, Tom Okello, Winnie Agwang, Joan Tusabe, Steven Ndugwa Kabwama, Anne Ruhweza Katahoire.

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
