## [Decision Letter · Decision Letter 0]

11 Jun 2022

PONE-D-21-39776Men's late presentation for HIV care in Eastern Uganda: The role of masculinity normsPLOS ONE

Dear Dr. Nabikande,

Thank you for submitting your manuscript to PLOS ONE. After careful consideration, we feel that it has merit but does not fully meet PLOS ONE’s publication criteria as it currently stands. Therefore, we invite you to submit a revised version of the manuscript that addresses the points raised during the review process.

Please see the comments from two reviewers below. The reviewers have asked for additional clarifications in the methodology employed. In particular, I note that both reviewers have queried the definitions used for masculinity and related concepts.

We look forward to receiving your revised manuscript.

Kind regards,

Hanna Landenmark

Staff Editor

PLOS ONE

Journal Requirements:

Reviewers' comments:

Reviewer's Responses to Questions

**Comments to the Author**

1. Is the manuscript technically sound, and do the data support the conclusions?

Reviewer #1: Yes

Reviewer #2: Partly

2. Has the statistical analysis been performed appropriately and rigorously? 

Reviewer #1: N/A

Reviewer #2: N/A

3. Have the authors made all data underlying the findings in their manuscript fully available?

Reviewer #1: Yes

Reviewer #2: Yes

4. Is the manuscript presented in an intelligible fashion and written in standard English?

Reviewer #1: Yes

Reviewer #2: Yes

5. Review Comments to the Author

Reviewer #1: Thank you, for inviting me to review this manuscript. Sherifah Nabikande1 et al thought to understanding why adult men present late for HIV care is important in improving early linkage to care and explored how masculinity norms influence men’s late presentation for HIV care. This is a very important study, I have few clarification comments

Abstract:

1. The engine of any study is the results: I think the results should be presented more. It is super summarized. Can it be expanded by including some demographic variables of the participants and other emerging themes, than only the three themes. Is it possible to relate the results to the theoretical framework used

2. Please, confirm that the three themes “fear of loss of respect, preservation of reputation, and maintaining a sense of normality means masculinity or equal to masculinity. I think the conclusion does not relate to the results improving linkage to care. Google define, masculinity as qualities or attributes regarded as characteristic of men. I am not sure the three norms really define, men’s masculinity

3. How do you define, masculinity? I think it would be good to define somewhere in the abstract

4. Under conclusion, please, add implication of the results and a recommendation (s)

Introduction

1. Line 70: respectively, statement “highlighted the influence of masculinity norms on men’s engagement in HIV care [11-13]” can you describe what masculinity norms of men in cite studies 11-13. Please, confirm to me that masculinity varies among men, because the study justification is that these studies were conducted among specific sub-populations whose social context did not necessarily reflect that of men in the general population irrespective of type of occupation

Methods

Under ethics, please, cite the study approval number, not the IRB USA FWA00011353. Did you register the study with UNCST? if yes, please, cite the UNCST number as well.

Discussion

Please, add strengthen and limitation of the study before conclusions.

Reviewer #2: This paper is interesting, and addresses an important topic. However the paper could be strengthened by a more in-depth discussion of the contextual aspects of masculinity in Uganda. Additionally, it would be helpful if the authors can discuss the implications of these findings, and make suggestions for how these barriers to men’s health seeking behaviour could be addressed.

- The introduction could benefit from more background on why early presentation of care is important

- First two sentences of introduction are a bit repetitive

- Line 63: comma unnecessary

- How is the theoretical framework relevant to the sociocultural context of Uganda? Wilson’s work is based on ethnographic work in the Caribbean in the 1960s. Discussions on masculinity should be situated and contextualised, geographically, socioculturally and temporally.

- What are the intersections between norms and expectations around masculinity and heteronormativity? There could be more reflection on this.

- What about the gender of research assistants who conducted the data collection? How would this have impacted on discussions around masculinity with participants?

- What kinds of questions were asked in IDIs and FGDs? How was masculinity probed?

- Line 161: were transcripts/translations reviewed and quality checked by someone other than the transcriber/translator to ensure correct interpretations?

- How did the analysis team collaborate? What kinds of discussions were there to ensure that there was consensus on coding and interpretation?

- The authors could strengthen discussion of stigma and shame. It is important to differentiate between the multiple dimensions of stigma that may serve as barriers towards seeking care, specifically for this study, highlight internalised stigma versus public stigma.

- Line 319: what is meant by the assertion that “society expects men to be wiser”?

- Similarly, what is meant by the statement that men are expected to “exercise self-control in sexual decision making”? How does this link to sexual scripts and norms around masculinity and heterosexuality in Uganda? What are the expectations around men’s sexual urges – could there be a link to masculinity expectations and an unbridled, untameable male sexual urge?

- Is there any link between masculinity and expectations of strength and vitality – which therefore links being HIV positive to associations of weakness?

- How can the framing of and messaging around ART help to address some of men’s concerns? Can the authors make any recommendations for this?

- Additionally, could there be any efforts to reframe health seeking behaviour as a sign of strength, looking after one’s health, that might help to shift men’s willingness to seek care?

6. PLOS authors have the option to publish the peer review history of their article (what does this mean?). If published, this will include your full peer review and any attached files.

Reviewer #1: No

Reviewer #2: No

---

## [Author Response · Author response to Decision Letter 0]

12 Aug 2022

REVIEWER 1

Comment 1: The engine of any study is the results: I think the results should be presented more. It is super summarized. Can it be expanded by including some demographic variables of the participants and other emerging themes, than only the three themes? Is it possible to relate the results to the theoretical framework used

Response: Dear reviewer, this has been improved as per your guidance as follows;

A total of 20 men participated in the In-depth Interviews (IDIs), with majority being married 15/20 (75%) and primary level holders 7/15 (46.7%). Nineteen (19) women participated in two FGDs, with a mean age of 29.5 years. Nineteen (19) men also participated in other two FGDs, with a mean age of 28.2 years. Conventional content analysis results indicated that men’s late presentation for HIV care in Jinja district is greatly related to their concerns of loss of respect and the need to preserve their reputation and maintain a sense of normality in their families and society as proposed by Wilson’s (1969) respectability-reputation theoretical model. Respectability was endorsed by ‘the wider society’, while reputation was endorsed almost entirely by men and some women.

Comment 2: Please, confirm that the three themes “fear of loss of respect, preservation of reputation, and maintaining a sense of normality means masculinity or equal to masculinity. I think the conclusion does not relate to the results improving linkage to care. 

Google define, masculinity as qualities or attributes regarded as characteristic of men. I am not sure the three norms really define, men’s masculinity

Response: Dear reviewer, the three themes; “fear of loss of respect, preservation of reputation, and maintaining a sense of normality highlight/reflect the attributes, roles, possessions or responsibilities that the men in the study setting do not want to lose because these are the socially enforced rules and expected roles and behavior associated with men and manhood in that culture.

This highlights the role of masculinity norms in men’s late presentation for HIV care in this context. This means that men did not present early for care because they did not want to lose the attributes, characteristics or qualities that makes them be viewed as men (masculinity) in this context.

Comment 3: How do you define, masculinity? I think it would be good to define somewhere in the abstract

Response: Masculinity is the social and cultural expression of what it means to a man (Kimmel 1987).

However, because there is diversity in contexts and experiences upon which notions of masculinity are constructed, the meaning of masculinity may vary from society to society (Connell and Messerschmidt 2005).

For this study, we were looking at masculinity norms defined as “socially enforced rules and expected roles and behavior associated with men and manhood in a given culture

Comment 4: Under conclusion, please, add implication of the results and a recommendation (s)

Response: Dear reviewer, this has been included as follows;

These findings have some implications for Public Health interventions. HIV programs that seek to encourage early HIV testing and treatment need to be more attentive to the values upheld my men associated with how their societies and cultures expect them to behave, the attributes and responsibilities they should have to be considered as men in that context. For example, Antiretroviral Therapy could be promoted by emphasizing its value in rebuilding a man’s respectability and role as head of family in regards to physical recovery. Supportive social spaces in which men can re-negotiate masculinities that are more aligned to their presentation for HIV care need to be created and men need to be supported with livelihood projects because this can help them provide for their families and encourage them to maintain HIV treatment. Further research is also needed to understand how masculinity norms can serve as a facilitator to men’s presentation for HIV care.

Comment 5: Line 70: respectively, statement “highlighted the influence of masculinity norms on men’s engagement in HIV care [11-13]” can you describe what masculinity norms of men in cite studies 11-13.

Response: Now studies [15-17]

In study (15), the masculine roles of worker and provider, husband and sexual partner, and the appearance of physical strength were compromised by HIV, but restored by ART’s positive effects on health. Occupational and structural factors specific to men’s role as worker remained as barriers to engagement in HIV care.

In study (16), Although HIV treatment improved most men’s health and enabled them to work again, in the process regaining self-worth, drug side-effects and disclosing HIV diagnosis and treatment to employers and work colleagues resulted in many men fearing to be, or actually being, judged as unable to work and being marginalized from work. This significantly undermined their masculine repute as hard workers, earners and providers for their families, leading many not to initiate HIV treatment or to drop out.

In study (17), men drew from a range of norms to fulfil the social and individual expectations of being sufficiently masculine. The study argues that there are essentially two forms of masculinity in Mam-Kiror, one based on reputation and the other on respectability, with some ideals shared by both. Respectability was endorsed by ‘the wider society’, while reputation was endorsed almost entirely by men.

Men’s treatment seeking behaviours corresponded with different masculine ideologies. Family and societal expectations to be a family provider and respectable role model encouraged treatment, to regain and maintain health.

However, reputational concern with strength and the capacity for hard physical work, income generation and sexual achievement discouraged uptake of HIV testing and treatment since it meant acknowledging weakness and an ‘HIV patient’ identity.

Comment 6: Please, confirm to me that masculinity varies among men, because the study justification is that these studies were conducted among specific sub-populations whose social context did not necessarily reflect that of men in the general population irrespective of type of occupation

Response: Dear reviewer, masculinities vary. The experience of masculinity is not uniform across or within cultural contexts. The central assertion here is that masculinity is dynamic, and enacted and sustained through and/or in different circumstances and as the society around a man changes through his lifetime. Hence it may vary according to the social groups of men that comprise a particular society.

Comment 7: Under ethics, please, cite the study approval number, not the IRB USA FWA00011353. Did you register the study with UNCST? If yes, please, cite the UNCST number as well.

Response: As a student, under Makerere University School of Public Health, we are required to seek approval from the School’s Research and Ethics Committee and that is the IRB number (FWA00011353) assigned to students.

In addition, we did not register the study with UNCST since it was not a requirement under the program.

Comment 8: Please, add strengthen and limitation of the study before conclusions.

Response: Study limitations and strengths have been added as follows;

This study adds to the growing body of research evidence that masculinity norms have a role in men’s access and engagement in HIV care. The generalizability of our findings is limited, given the qualitative nature of the study, as well as the use of purposive sampling to recruit participants. In addition, In Depth Interview (IDI) questions were based on past experiences and this might have resulted in recall bias. However, this was minimized by including participants who had presented for HIV care in recent months/years (between January 2018 and June 2020).

REVIEWER 2

Comment 1: This paper is interesting, and addresses an important topic. However the paper could be strengthened by a more in-depth discussion of the contextual aspects of masculinity in Uganda.

Response: Dear reviewer, thank you for this comment. We have added a more in-depth discussion of the contextual aspects of masculinity in Uganda under the materials and methods.

Uganda is ethnically diverse, with over 30 different ethnic groups contained within its borders. Masculinity remains strongly tied to the breadwinner identity and one’s role and way of participating in family and community life continues to be highly designated by one’s sex. Men hold more power and status in both family and community spheres and are considered responsible for productive labor outside of the home. In addition, there exists one of the dominant masculine ideals of being a provider, primarily embodied through responsible fatherhood.

Comment 2: Additionally, it would be helpful if the authors can discuss the implications of these findings, and make suggestions for how these barriers to men’s health seeking behaviour could be addressed.

Response: Dear reviewer, this has been included under the conclusion.

These findings have some implications for Public Health interventions. HIV programs that seek to encourage early HIV testing and treatment need to be more attentive to the values upheld my men associated with how their societies and cultures expect them to behave, the attributes and responsibilities they should have to be considered as men in that context. For example, Antiretroviral Therapy could be promoted by emphasizing its value in rebuilding a man’s respectability and role as head of family in regards to physical recovery. Supportive social spaces in which men can re-negotiate masculinities that are more aligned to their presentation for HIV care need to be created and men need to be supported with livelihood projects because this can help them provide for their families and encourage them to maintain HIV treatment. Further research is also needed to understand how masculinity norms can serve as a facilitator to men’s presentation for HIV care.

Comment 3: The introduction could benefit from more background on why early presentation of care is important

Response: We appreciate this input and have included this;

Early presentation for care is important for reducing the risk of mortality and improving quality of life among people with HIV [9]. It can also reduce the risks of virological failure, and loss to care (attrition) [10-12]

Comment 4: First two sentences of introduction are a bit repetitive

Response: Dear reviewer, thank you for bringing this to our attention. The second sentence has been removed.

Comment 5: Line 63: comma unnecessary

Response: This has been rectified

Comment 6: How is the theoretical framework relevant to the sociocultural context of Uganda? Wilson’s work is based on ethnographic work in the Caribbean in the 1960s. Discussions on masculinity should be situated and contextualised, geographically, socioculturally and temporally.

Response: Dear reviewer, although Wilson’s work was based on ethnographic work in the Caribbean in the 1960s, there are several insightful examples of their application in research among Caribbean and the African American communities, which can be used to explain the experiences of being a man in Uganda in the context of HIV treatment. For example the constructs that were central for establishing respectability and reputation such as having multiple sexual partners, responsible fatherhood and provider roles, are common dimensions of masculinity even in the Ugandan setting.

In addition, this theory has previously been utilized by other researchers in the Ugandan setting. For example; (Siu, Seeley, and Wight 2013; Mbonye, Siu & Seeley 2022) 

Comment 7: What are the intersections between norms and expectations around masculinity and heteronormativity? There could be more reflection on this.

Response: Thank you for bringing this up. Unfortunately, for this study, we did not explore these intersections. However, this is an important subject that we would like to explore in future.

Comment 8: What about the gender of research assistants who conducted the data collection? How would this have impacted on discussions around masculinity with participants?

Response: Four (4) research assistants (RA); of which 3 were male and 1 was female, conducted the data collection. Participants might not have freely and fully opened up about issues surrounding masculinity to the female research assistant.

However, this limitation was catered for by making sure more interviews/discussions were conducted by the male research assistants compared to those conducted by the female research assistant.

Note: all FGDs were conducted by a male RA and out of the 20 IDIs, only 5 were conducted by the female RA.

Comment 9: What kinds of questions were asked in IDIs and FGDs? How was masculinity probed?

Response: Dear reviewer, the questions asked in the IDIs and FGDs included;

1. What are the things that a man has to do to be considered a man in this region/district?

2. What are some of the most desirable masculine traits in your community?

3. How is HIV/AIDs perceived in this region? Do you think being HIV positive changes how a man is viewed in this region? (Probe: answers with examples (By his friends, his family, himself if yes, how?)

4. How do the beliefs and definition of what it means to be a man in this region influence men’s presentation for HIV care? (Probe: is the influence positive/negative? do men find it easy to seek HIV treatment, do they find it difficult to start HIV treatment?)

5. How about you? Did your beliefs of what it means to be a man influence your access/presentation for HIV care? (Probe: how did they influence your decision to present into HIV care, was it easy/difficult for you? Explain/How?)

6. How about you? Did your beliefs of what it means to be a man influence your access/presentation for HIV care? (Probe: how did they influence your decision to present for HIV care, was it easy/difficult for you? Explain/How?)

7. Why do some men appear to have greater difficulties or less motivation to present early for HIV care and how do others find it easy to do so? (Probe: Are there aspects of masculinity that make it easy for men to seek help? which ones? Why? Are there aspects of masculinity that make it difficult for men to seek help? Which ones? Why?)

Comment 10: Line 161: were transcripts/translations reviewed and quality checked by someone other than the transcriber/translator to ensure correct interpretations?

Response: New line 212-214

Yes. Since recruited Research Assistants were fluent in both Lusoga and English, translated transcripts were distributed among them to ensure correct interpretations. 

The manuscript has been updated with this information.

Comment 11: How did the analysis team collaborate? What kinds of discussions were there to ensure that there was consensus on coding and interpretation?

Response: The analysis team comprised of three people (S.N, W.A, J.T) and these collaborated by holding physical team meetings throughout the analysis process.

A week-by-week plan for analysis was drafted. The weekly plan included a reference to the data analysis phase and coding assignments for each team member.

During each iteration of coding, team members coded the same two to three transcripts into the codebook. Then, during research team meetings, each coded statement was compared across members of the analysis team. Disagreements were discussed until the group reached consensus.

Much of the discussions revolved around the similarities, differences, connections and patterns within and between the codes. The discussions were facilitated by (A.R.K) and these revealed high levels of agreement between the team, and a consensus was able to be reached about the data. This synthesis was enabled by careful listening, lengthy and full discussion, a joint reflection on shared codes and ideas and an interweaving of our interpretations.

After all of the transcripts had been coded using consensus coding, the research team then met one final time to review the codebook. During the meeting, the codebook was developed into a thematic structure that comprised of categories and associated themes that described the role of masculinity norms in men’s presentation for HIV care. The themes were then reviewed and approved by all members of the analysis team.

Comment 12: The authors could strengthen discussion of stigma and shame. It is important to differentiate between the multiple dimensions of stigma that may serve as barriers towards seeking care, specifically for this study, highlight internalised stigma versus public stigma.

Response: Dear reviewer, we appreciate this input and we have included this suggestion in the manuscript as follows;

Different forms of stigma have been implicated as barriers to achieving the full potential of numerous clinical advances to improve health at the population level, including ART treatment as prevention [22-24]. Internalised stigma has been found to impact certain decisions like avoiding to seek care when needed [25]. Several studies have also shown an association between internal stigma and poor affective health and well-being as well as poor health care behaviors [26-27]. Internalised HIV stigma encompasses feelings of being “less than” others, dirty or unclean, and/or deserving of negative outcomes due to having HIV [27]. On the other hand, “enacted stigma,” which is the lived experience of discrimination against an HIV positive person through others’ actions or words, is also known to have a negative impact on health outcomes [28].

Comment 13: Line 319: what is meant by the assertion that “society expects men to be wiser”?

Response: Dear reviewer, the full statement is that “society expects men to be wiser and exercise self-control in sexual decision making.” This means that since society views men to be the more dominant gender compared to women, they are expected to make safer sexual choices and practice safe sex so as not to contract HIV. Acquiring HIV is interpreted as a failure in the man’s sense of judgement in regards to sexual decision making.

Comment 14: Similarly, what is meant by the statement that men are expected to “exercise self-control in sexual decision making”? How does this link to sexual scripts and norms around masculinity and heterosexuality in Uganda? 

Response: Society expecting men to exercise self-control in sexual decision making is linked to sexual scripts and norms around masculinity and heterosexuality in Uganda in a number of ways.

Although there is variation between cultures, traditional gender roles generally portray men as the dominant, controlling initiators in heterosexual relationships. The culturally scripted expectation that women should defer to a male partner’s sexual needs gives primacy to male pleasure at the expense of safer-sex measures. Because of this, women tend to leave condom initiation and use to men and also accept that men can have multiple sexual partners.

It’s upon this background that we argue as authors, that since men have the upper hand in regards to sexual decision making, they have the ability to exercise self-control and engage in safer sexual encounters.

Comment 15: What are the expectations around men’s sexual urges – could there be a link to masculinity expectations and an unbridled, untameable male sexual urge?

Response: Sexual urge refers to sex drive, or libido, and is defined as a person's desire to engage in sexual activity.

As masculinity is constructed through discourse and interactions, the discourse surrounding the ‘male sexual drive’ propagates the idea that men are biologically programmed to constantly and relentlessly desire sex (Holloway, 1984; Holloway, 1996). 

The ‘male sexual drive’ has its roots in historical notions that men need to ‘spread their seed’ and are ‘hardwired’ to have unprotected sex with multiple women partners to reproduce many offspring (Gilmore, 1990; Thornhill & Palmer, 2001; Hagen, 1979). While these biological explanations have largely been discredited (Hunter, 2005; Fine’ 2010), the concept of men’s sexuality being biologically hardwired remains pervasive in popular culture and discourse (Coyne, 2000). Because of this perceived sex drive, some men describe themselves as acting on instincts which did not allow them to abstain or to use condoms during sexual intercourse. 

Comment 16: Is there any link between masculinity and expectations of strength and vitality – which therefore links being HIV positive to associations of weakness?

Response: Dear reviewer, characteristics of vitality and strength are commonly ascribed to masculinity and can be required in certain contexts to achieve masculine status. While there are a variety of behaviors men can use to demonstrate virility, vitality and strength, men sometimes use their sexual activity, capacity for drinking, or shows of force to demonstrate these masculine characteristics for their peers (Courtenay 2000, Hatcher et al. 2014).

In this way, some men’s behaviors, including behaviors known to be risk factors for HIV, may help them construct an outward image aligned with the dominant ideal masculinity but can negatively affect their health.

Being HIV positive is not linked to associations of weakness but the consequences of HIV such as body weakness, regular sickness threaten men’s ability to demonstrate these masculine characteristics of vitality and strength.

Comment 17: How can the framing of and messaging around ART help to address some of men’s concerns? Can the authors make any recommendations for this?

Response: Dear reviewer, thank you for bringing this up. Framing of and messaging around ART by emphasizing that ART restores physical health. 

For example; physical recovery means strength, and so an ability to work again, which means men are able to provide for their families as men are expected to, which brings us back respectability.

This has been included in the manuscript.

Comment 18: Additionally, could there be any efforts to reframe health seeking behaviour as a sign of strength, looking after one’s health, that might help to shift men’s willingness to seek care?

Response: Dear reviewer, we believe these efforts could be fruitful by health workers emphasizing how health seeking can maintain or improve characteristics especially the physical ones that are key for men in terms of proving their masculinity. This has been included in the manuscript.

References not in the manuscript used to answer some of the comments;

1. Courtenay WH. Constructions of masculinity and their influence on men’s well-being: a theory of gender and health. Social Science & Medicine. 2000; 50(10):1385–1401. Epub 2000/03/31. [PubMed: 10741575]

2. Hatcher AM, Colvin CJ, Ndlovu N, Dworkin SL. Intimate partner violence among rural South African men: alcohol use, sexual decision-making, and partner communication. Culture, Health & Sexuality. 2014; 16(9):1023–1039. Epub 2014/06/19.

3. Holloway W. Gender difference and the production of subjectivity. In: Henriques J, Holloway W, Urwin C, Venn C, Walkerdine V, editors. Changing the subject: Psychology, Social regulation and subjectivity. London: Methuen; 1984. 

4. Holloway W. In: Theorizing Heterosexuality Recognition and heterosexual desire. Richardson D, editor. Buckinghamshire, UK: Open University Press; 1996

5. Gilmore DD. Manhood in the Making: Cultural Concepts of Masculinity. New Haven: Yale University Press; 1990

6. Thornhill R, Palmer C. A natural history of rape: Biological bases of sexual coercion. Cambridge, MA: MIT Press; 2001.

7. Hagen R. The bio-sexual factor. New York, NY: Doubleday; 1979

8. Hunter M. Cultural politics and masculinities: multiple-partners in historical perspective in KwaZulu-Natal. Cult Health Sex. 2005;7(3):209–223. Epub 2006/08/01

9. Fine C. Delusions of gender: How our minds, society, and neurosexism create difference. WW Norton & Company; 2010

10. Coyne JA. Of vice and men: The fairy tales of evolutionary psychology. New Republic. 2000;147:27–34.

---

## [Decision Letter · Decision Letter 1]

29 Aug 2022

PONE-D-21-39776R1Men's late presentation for HIV care in Eastern Uganda: The role of masculinity normsPLOS ONE

Dear Dr. Nabikande,

Thank you for submitting your manuscript to PLOS ONE. After careful consideration, we feel that it has merit but does not fully meet PLOS ONE’s publication criteria as it currently stands. Therefore, we invite you to submit a revised version of the manuscript that addresses the points raised during the review process.

We look forward to receiving your revised manuscript.

Kind regards,

Nelsensius Klau Fauk, S.Fil., M., MHID, MSc, PhD

Academic Editor

PLOS ONE

Journal Requirements:

Additional Editor Comments:

Dear Authors,

Thanks for submitting the revised version of the manuscript. You have addressed the all the comments from the reviewers and improved the manuscript.

I have a few minor comments for you to address and suggestions to support some of your statements with some resources that I provided. Please quickly address these and your manuscript will be finally accepted.

Overall comments to authors:

The use terms “HIV positive men” is stigmatising by WHO, please change and use terms like “men living with HIV” or “people living with HIV” or “patients living with HIV” throughout the manuscript.

Lines 75-76: “Although women in most parts of Sub-Saharan Africa (SSA), and Uganda in particular, have a higher prevalence of HIV than men”

Please use this source to support it:

• B Sartorius, JD VanderHeide, M Yang, EA Goosmann, et al. Subnational mapping of HIV incidence and mortality among individuals aged 15–49 years in sub-Saharan Africa, 2000–18: a modelling study. The Lancet HIV 8 (6), e363-e375. https://www.sciencedirect.com/science/article/pii/S2352301821000515

Lines 80-81: “Fewer men seek testing and counselling services on a routine basis leading to late diagnosis in late-stage disease progression”

Please use the following sources to support it:

• The intention of men who have sex with men to participate in voluntary counseling and HIV testing and access free condoms in Indonesia. American Journal of Men's Health 12 (5), 1175-1184. https://journals.sagepub.com/doi/full/10.1177/1557988318779737

• Perceptions among transgender women of factors associated with the access to HIV/AIDS-related health services in Yogyakarta, Indonesia. PloS one 14 (8), e0221013. https://journals.plos.org/plosone/article?id=10.1371/journal.pone.0221013

• Risk Factors and the Impact of HIV among Women Living with HIV and Their Families in Yogyakarta and Belu District, Indonesia. ( see Pages 35-54: scoping review on barriers and facilitators of the access of men living with HIV to HIV services; pages 224-247: findings on barriers and facilitators of the access of men living with HIV to HIV services). https://flex.flinders.edu.au/file/eac1a04e-a24b-4888-ace8-50f06805069e/1/Fauk%20Thesis2022.pdf

Lines 82-83: Due to late presentation, men develop opportunistic infections with poor treatment outcomes [7] and high healthcare costs [REF].

Please use the following source to support ‘high healthcare costs:

• Risk Factors and the Impact of HIV among Women Living with HIV and Their Families in Yogyakarta and Belu District, Indonesia. ( see Pages 35-54: scoping review on barriers and facilitators of the access of men living with HIV to HIV services; pages 224-247: findings on barriers and facilitators of the access of men living with HIV to HIV services). https://flex.flinders.edu.au/file/eac1a04e-a24b-4888-ace8-50f06805069e/1/Fauk%20Thesis2022.pdf

Data management and analysis

Are the field researchers (research assistants) the authors in the paper? If yes, please put their initials to make clear who did what. Who are the three people in the analysis team? Please provide their initials in bracket.

I suggest the authors use the following resources to support their description of data analysis:

• Cultural and religious determinants of HIV transmission: A qualitative study with people living with HIV in Belu and Yogyakarta, Indonesia. PLoS ONE 16(11): e0257906. https://journals.plos.org/plosone/article?id=10.1371/journal.pone.0257906

• Qualitative Data Analysis for Applied Policy Research. In: Bryman A, Burgess RG, editors. London: Routledge; 1994. p. 173–94.

Discussion

Lines 389-390: “Internalised stigma has been found to impact certain decisions like avoiding to seek care when needed.”

Please use the following resources to support it:

• HIV Stigma and Moral Judgement: Qualitative Exploration of the Experiences of HIV Stigma and Discrimination among Married Men Living with HIV in Yogyakarta. Int. J. Environ. Res. Public Health 2020, 17, 636. https://doi.org/10.3390/ijerph17020636.

• HIV stigma and discrimination: perspectives and personal experiences of healthcare providers in Yogyakarta and Belu, Indonesia. Frontiers in medicine 8, 625787. https://www.frontiersin.org/articles/10.3389/fmed.2021.625787/full

Lines 394-396: “On the other hand, “enacted stigma,” which is the lived experience of discrimination against an HIV positive person through others’ actions or words, is also known to have a negative impact on health outcomes.”

Please use the following resources to support it:

• HIV Stigma and Moral Judgement: Qualitative Exploration of the Experiences of HIV Stigma and Discrimination among Married Men Living with HIV in Yogyakarta. Int. J. Environ. Res. Public Health 2020, 17, 636. https://doi.org/10.3390/ijerph17020636.

• Stigma and Discrimination towards People Living with HIV in the Context of Families, Communities, and Healthcare Settings: A Qualitative Study in Indonesia. Int. J. Environ. Res. Public Health 2021, 18, 5424. https://doi.org/10.3390/ijerph18105424

Reviewers' comments:

Reviewer's Responses to Questions

**Comments to the Author**

1. If the authors have adequately addressed your comments raised in a previous round of review and you feel that this manuscript is now acceptable for publication, you may indicate that here to bypass the “Comments to the Author” section, enter your conflict of interest statement in the “Confidential to Editor” section, and submit your "Accept" recommendation.

Reviewer #2: All comments have been addressed

2. Is the manuscript technically sound, and do the data support the conclusions?

Reviewer #2: Yes

3. Has the statistical analysis been performed appropriately and rigorously? 

Reviewer #2: N/A

4. Have the authors made all data underlying the findings in their manuscript fully available?

Reviewer #2: Yes

5. Is the manuscript presented in an intelligible fashion and written in standard English?

Reviewer #2: Yes

6. Review Comments to the Author

Reviewer #2: The authors have done a good job at addressing reviewer comments. The paper is much improved. My only comment is that some of the author responses to reviewer comments could have been added into the paper too - for example:

Reviewer 2: comments 6, 7, 8, 13, 14, 15, and 16. The authors' responses to reviewer comments would also strengthen the paper and demonstrate the authors' engagement with these questions and concepts.

7. PLOS authors have the option to publish the peer review history of their article (what does this mean?). If published, this will include your full peer review and any attached files.

Reviewer #2: No

---

## [Author Response · Author response to Decision Letter 1]

27 Oct 2022

The use terms “HIV positive men” is stigmatising by WHO, please change and use terms like “men living with HIV” or “people living with HIV” or “patients living with HIV” throughout the manuscript.

Response: Dear reviewer, this has been rectified as per your guidance to “men living with HIV.”

Lines 75-76: “Although women in most parts of Sub-Saharan Africa (SSA), and Uganda in particular, have a higher prevalence of HIV than men”

Please use this source to support it:

• B Sartorius, JD VanderHeide, M Yang, EA Goosmann, et al. Subnational mapping of HIV incidence and mortality among individuals aged 15–49 years in sub-Saharan Africa, 2000–18: a modelling study. The Lancet HIV 8 (6), e363-e375. https://www.sciencedirect.com/science/article/pii/S2352301821000515

Response: Dear reviewer, we appreciate this suggested resource. Unfortunately, findings from this source were not stratified by sex. It is therefore impossible to know which sex had a higher HIV prevalence

Lines 80-81: “Fewer men seek testing and counselling services on a routine basis leading to late diagnosis in late-stage disease progression”

Please use the following sources to support it:

• The intention of men who have sex with men to participate in voluntary counseling and HIV testing and access free condoms in Indonesia. American Journal of Men's Health 12 (5), 1175-1184. https://journals.sagepub.com/doi/full/10.1177/1557988318779737

• Perceptions among transgender women of factors associated with the access to HIV/AIDS-related health services in Yogyakarta, Indonesia. PloS one 14 (8), e0221013. https://journals.plos.org/plosone/article?id=10.1371/journal.pone.0221013

• Risk Factors and the Impact of HIV among Women Living with HIV and Their Families in Yogyakarta and Belu District, Indonesia. ( see Pages 35-54: scoping review on barriers and facilitators of the access of men living with HIV to HIV services; pages 224-247: findings on barriers and facilitators of the access of men living with HIV to HIV services). https://flex.flinders.edu.au/file/eac1a04e-a24b-4888-ace8-50f06805069e/1/Fauk%20Thesis2022.pdf

Response: Dear reviewer, we appreciate these recommended sources. We have reviewed them and used the resources to support the statement as follows;

Low perceptions or knowledge about HIV/AIDS and the benefits of Voluntary Counseling and Testing (VCT) services have been reported for men’s negative attitudes to access HIV/AIDS services [7]. The provision of information can improve individuals’ HIV-related health literacy [8]. Health literacy is required in making critical health decisions to manage individuals’ health, including to seek and access appropriate health care [9]. In addition, men’s supportive social relationships with the healthcare professionals, their ease of access to the HIV care services, and their ability to seek the services have been reported to determine their access to the services [10].

Lines 82-83: Due to late presentation, men develop opportunistic infections with poor treatment outcomes [7] and high healthcare costs [REF].

Please use the following source to support ‘high healthcare costs:

• Risk Factors and the Impact of HIV among Women Living with HIV and Their Families in Yogyakarta and Belu District, Indonesia. ( see Pages 35-54: scoping review on barriers and facilitators of the access of men living with HIV to HIV services; pages 224-247: findings on barriers and facilitators of the access of men living with HIV to HIV services). https://flex.flinders.edu.au/file/eac1a04e-a24b-4888-ace8-50f06805069e/1/Fauk%20Thesis2022.pdf

Response: Dear reviewer, we appreciate these recommended sources. We have reviewed them and used the resources to support the statement as follows;

The costs of medical treatment, such as the costs for standard blood, viral load and CD4 tests prior to the initiation of treatment, the treatment of opportunistic infections and administration have also been revealed to be barriers to the access of HIV care services among People Living with HIV (PLHIV) [10, 12, 13, 14]

Data management and analysis

Are the field researchers (research assistants) the authors in the paper? If yes, please put their initials to make clear who did what. Who are the three people in the analysis team? Please provide their initials in bracket.

Response: Dear reviewer, the research assistants are not part of the authors. They were recruited, and trained to do data collection. 

The analysis team comprised of three people (S.N, W.A, J.T)

I suggest the authors use the following resources to support their description of data analysis:

• Cultural and religious determinants of HIV transmission: A qualitative study with people living with HIV in Belu and Yogyakarta, Indonesia. PLoS ONE 16(11): e0257906. https://journals.plos.org/plosone/article?id=10.1371/journal.pone.0257906

• Qualitative Data Analysis for Applied Policy Research. In: Bryman A, Burgess RG, editors. London: Routledge; 1994. p. 173–94.

Response: We appreciate this suggestion. These resources have supported us to strengthen the description of our data analysis as follows;

Data analysis was guided by the five steps of qualitative data analysis framework [28]. These include (i) familiarization with the data by reading through transcripts, listening to the audios and going through the field notes; (ii) identification of a thematic framework by making notes, writing down recurrent themes and issues which emerge as important to study respondents; (iii) indexing the data by applying the thematic framework to the individual transcripts, grouping similar codes under the same themes and sub-themes; (iv) charting data through arranging appropriate thematic references in a summary of chart; and (v) mapping and interpretation of the data through which data were examined and interpreted [28-29]. 

Discussion

Lines 389-390: “Internalised stigma has been found to impact certain decisions like avoiding to seek care when needed.”

Please use the following resources to support it:

• HIV Stigma and Moral Judgement: Qualitative Exploration of the Experiences of HIV Stigma and Discrimination among Married Men Living with HIV in Yogyakarta. Int. J. Environ. Res. Public Health 2020, 17, 636. https://doi.org/10.3390/ijerph17020636.

• HIV stigma and discrimination: perspectives and personal experiences of healthcare providers in Yogyakarta and Belu, Indonesia. Frontiers in medicine 8, 625787. https://www.frontiersin.org/articles/10.3389/fmed.2021.625787/full

Response: Dear reviewer, we appreciate this suggestion. These resources have supported us to strengthen the statement as follows;

A study in Indonesia reported that participants’ individual moral judgement that associated their HIV status with amoral behaviours and engagement in sex with female sex workers, and participants’ self-judgement as ‘dirty people’ or sinners which increased guilty feelings, were additional factors leading to HIV’s perceived stigma [37]

Lines 394-396: “On the other hand, “enacted stigma,” which is the lived experience of discrimination against an HIV positive person through others’ actions or words, is also known to have a negative impact on health outcomes.”

Please use the following resources to support it:

• HIV Stigma and Moral Judgement: Qualitative Exploration of the Experiences of HIV Stigma and Discrimination among Married Men Living with HIV in Yogyakarta. Int. J. Environ. Res. Public Health 2020, 17, 636. https://doi.org/10.3390/ijerph17020636.

• Stigma and Discrimination towards People Living with HIV in the Context of Families, Communities, and Healthcare Settings: A Qualitative Study in Indonesia. Int. J. Environ. Res. Public Health 2021, 18, 5424. https://doi.org/10.3390/ijerph18105424

Response: Dear reviewer, we appreciate this suggestion. These resources have supported us to strengthen the statement as follows;

Individuals have revealed having experienced HIV-related discrimination attitudes from healthcare professionals within healthcare settings. For example, in the Indonesia study, it was reported that clients were left untreated for many days in healthcare facilities [37]. HIV stigma and discrimination by healthcare professionals traumatized the PLHIV, thus concealment of HIV status, hindering access to healthcare services [37, 39, 40] thereby influencing their health and wellbeing in the long run.

---

## [Editor Report · Decision Letter 2]

31 Oct 2022

Men's late presentation for HIV care in Eastern Uganda: The role of masculinity norms

PONE-D-21-39776R2

Dear Dr. Nabikande,

We’re pleased to inform you that your manuscript has been judged scientifically suitable for publication and will be formally accepted for publication once it meets all outstanding technical requirements.

Kind regards,

Nelsensius Klau Fauk, S.Fil., M., MHID, MSc, PhD

Academic Editor

PLOS ONE
---

## [Editor Report · Acceptance letter]

8 Nov 2022

PONE-D-21-39776R2 

Men’s late presentation for HIV care in Eastern Uganda: The role of Masculinity Norms 

Dear Dr. Nabikande:

I'm pleased to inform you that your manuscript has been deemed suitable for publication in PLOS ONE. Congratulations! Your manuscript is now with our production department. 

Kind regards, 

on behalf of

Dr. Nelsensius Klau Fauk 

Academic Editor

PLOS ONE